

# Exploring coral reef responses to millennial scale climatic forcings: insights from a 1-D numerical tool pyReef-Core v1.0

Tristan Salles[1], Jodie Pall[1], M. Jody Webster[1], and Belinda Dechnik[2]

[1]Geocoastal Research Group, School of Geosciences, University of Sydney, Sydney, NSW, 2006, Australia
[2]Department of Oceanography and Ecology, Federal University of Espirito Santo, Vitoria, ES CEP-29075-910, Brazil

*Correspondence to:* Tristan Salles (tristan.salles@sydney.edu.au)

**Abstract.**

Assemblages of corals characterise specific reef biozones and the environmental conditions that change laterally across a reef and with depth. Drill cores through fossil reefs record the time- and depth-distribution of assemblages, which captures a partial history of the vertical growth response of reefs to changing palaeoenvironmental conditions. The effects of environmental factors on reef growth are well understood on ecological time-scales but are poorly constrained at centennial to millennial timescales. *pyReef-Core* is a stratigraphic forward model designed to solve the inverse problem of unobservable environmental processes controlling vertical reef development by simulating the physical, biological and sedimentological processes that determine vertical assemblage changes in drill cores. It models the stratigraphic development of coral reefs at centennial to millennial timescales under environmental forcing conditions including accommodation (relative sea level upward growth), oceanic variability (flow speed, nutrients, pH and temperature), sediment input and tectonics. It also simulates competitive coral assemblage interactions using the generalised Lotka-Volterra system of equations (GLVEs) and can be used to infer the influence of environmental conditions on the zonation and vertical accretion and stratigraphic succession of coral assemblages over decadal timescales and greater. The tool can quantitatively test carbonate platform development under the influence of ecological and environmental processes, and efficiently interpret vertical growth and karstification patterns observed in drill cores. We provide two realistic case studies illustrating the basic capabilities of the model and use it to reconstruct (1) the Holocene history (from 8,500 years to present) of coral community responses to environmental changes, and (2) the evolution of an idealised coral-reef core since the Last Interglacial (from 140,000 years to present) under the influence of sea-level change, subsidence and karstification. We find that the model reproduces the details of the formation of existing coral-reef stratigraphic sequences both in terms of assemblages succession, accretion rates and depositional thicknesses. It can be applied to estimate the impact of changing environmental conditions on growth rates and patterns under many different settings and initial conditions.





## 1 Introduction

Ecologists and geologists tend to have different spatial and temporal perspectives of coral reefs. This is because the methods and observations which inform both fields differ. While ecologists can make direct oceanographic and biological observations of coral reef ecosystems on daily to decadal timescales, reef geologists must interpret assemblage patterns from fossil outcrops and drill cores to infer persistent biological or sedimentological processes on centennial to millenial timescales. This results in both fields addressing differently the question of how coral reefs respond to environmental conditions (Stocker et al., 2013). Furthermore, as Hughes (2000) argues, the most relevant spatial and temporal scales fall in the gap between both fields; modelling predictions of climate change are most relevant to society on regional to global scales over hundreds of years.

Stratigraphic forward modelling (SFM) of carbonate systems offer a solution by simulating sedimentary processes and carbonate production through time (Burgess and Wright, 2003). In this paper, we present a deterministic, one-dimensional (1-D) numerical model, **pyReef-Core**, that simulates the vertical coral growth patterns observed in a drill core, as well as the physical and environmental processes that affect coral growth. The model is capable of integrating ecological processes like coral community interactions over centennial-to-millennial scales using predator-prey or *Generalised Lotka-Volterra Equations* (GLVEs). *pyReef-Core* is the first of its kind to incorporate coral community dynamics into reef growth modelling at reef-scale resolution. We first describe the main model constitutive laws and forcing parameters. Then we present two realistic case studies to illustrate the model's capability. First, we simulate a Holocene shallowing-up fossil reef sequence representing a 'catch-up' growth strategy observed in the Great Barrier Reef (Hopley et al., 2007; Dechnik et al., 2017) and estimate assemblage compositions and changes. The second case study simulates the long-term evolution ($> 120,000$ years) of an idealised reef sequence under the influence of sea-level change and subsidence, commonly observed on passive margins world wide (Montaggioni, 2005; Woodroffe and Webster, 2014; Gischler, 2015).

## 2 SFM of carbonate systems

SFM has become a powerful tool used to predict stratigraphic architecture of sedimentary systems (Warrlich et al., 2008). SFM involves simulating processes acting over geological timescales, and iteratively refining parameters to improve the match between observed and predicted morphologies and stratigraphies. Through this trial-and-error procedure, parameters such as sedimentation and carbonate production rates can be evaluated and quantified, where they ordinarily cannot be directly observed from the fossil record (Dalmasso et al., 2001; Warrlich et al., 2008; Salles et al., 2011; Seard et al., 2013; Huang et al., 2015). In that sense, SFM addresses the short-comings of qualitative investigation techniques applied to carbonate systems (e.g., Cabioch et al., 1999; Abbey et al., 2011; Dechnik et al., 2015). Several numerical models have been developed since the 1960s to investigate the evolution of carbonate systems, yet only recently have the complexity of biological interactions – specific to reefs – started to be addressed (Barrett and Webster, 2017; Clavera-Gispert et al., 2017).

Traditionally stratigraphic modelling of carbonate-siliciclastic systems has been applied to locate oil and gas reservoirs (Kendall et al., 1991; Burgess et al., 2006; Warrlich et al., 2008; Hill et al., 2009). However, SFM has become a popular heuristic tool to better understand and quantify parameters regulating peritidal carbonates (Burgess and Prince, 2015), the development of





coral reef environments (Bosscher and Southam, 1992; Clavera-Gispert et al., 2017) as well as microbial (Parcell, 2003) and coral reef growth (Paulay and McEdward, 1990; Bosscher and Southam, 1992; Dalmasso et al., 2001). Early forward models were 1-D (Schwarzacher, 1966) or 2-D in formulation (Bosence and Waltham, 1990; Kendall et al., 1991), but improvements in computing led to the development of more complex, 3-D models (e.g., DIONISOS (Granjeon and Joseph, 1999; Seard et al.,

2013) and FUZZIM (Nordlund, 1999)).

Most recently, three software packages have been developed that represent important antecedents to the modelling effort described in this paper: CARBONATE-3D (C3D) (Warrlich et al., 2008), ReefSAM (Barrett and Webster, 2017), and SIMSAFADIM-CLASTIC (Clavera-Gispert et al., 2017). These models are 3-D and able to simulate hydrodynamic processes, sediment transport and biological production, but with varying degrees of realism. ReefSAM and C3D are both reef-scale models, yet ReefSAM

constitutes an improvement from C3D in prediction of more realistic reef growth morphologies (i.e. lagoonal patch reefs and mostly sand infilled lagoons) that depends on environmental factors (Barrett and Webster, 2017). However, despite the added complexity, ReefSAM, like C3D, was found to have overly simplistic hydrodynamic and sediment transport models that were unable to simulate important, small-scale morphological features and feedbacks (Barrett and Webster, 2017).

The shortcomings of both textscReefSAM and C3D are notable in their inability to model bio-sedimentary facies in any com-

plexity. Limited to basic sedimentary facies only, they also fail to simulate how changing environmental conditions influence the ecological requirements of different coral reef communities (Clavera-Gispert et al., 2017). SIMSAFADIM-CLASTIC offers the possibility to investigate carbonate production as a biological function of species interactions (based on the Lotka-Volterra equations) as well as environmental parameters (i.e., light, hydrodynamic energy and slope) (Clavera-Gispert et al., 2017). However, it has only been applied to model interactions between marine organisms and not between reef building corals. Fur-

thermore, while the approach is promising, SIMSAFADIM-CLASTIC is not applicable at reef-scales due to its coarse > 100 m spatial resolution and with a minimum time interval exceeding the life-span of corals (500 years).

3-D SFM becomes necessary when accounting for the 3-D nature of sediment-driven and hydrodynamic processes like lateral reef accretion and fluid flow, establishing sediment budgets or investigating problems such as the influence of inherited topography (Warrlich et al., 2008). However, the development of complex 3-D models has not necessarily improved the quality

of carbonate system modelling. In some cases lower-dimensional and reduced-complexity models are easier to test and constrain (Paola, 2000). Because 1-D forward modelling prioritises accommodation space as the fundamental control over vertical sequences, it is a starting point to understand and constrain other essential influences on reef growth before adding greater complexity. Rationalised this way, *pyReef-Core* serves as a basis for constraining the biological interactive aspect of carbonate production, and the effect of environmental influences. Once an understanding of the complex influence of environmental

conditions on vertical coral accretion can be established, extending the model to 2-D and 3-D becomes a less challenging task.

## 3    Environmental controls on reef development

Coral framework production is strongly linked to biological activity, such that the evolution of reef systems are limited by the growth potential of carbonate producing organisms and their environmental requirements (Flügel, 2004). Environmental





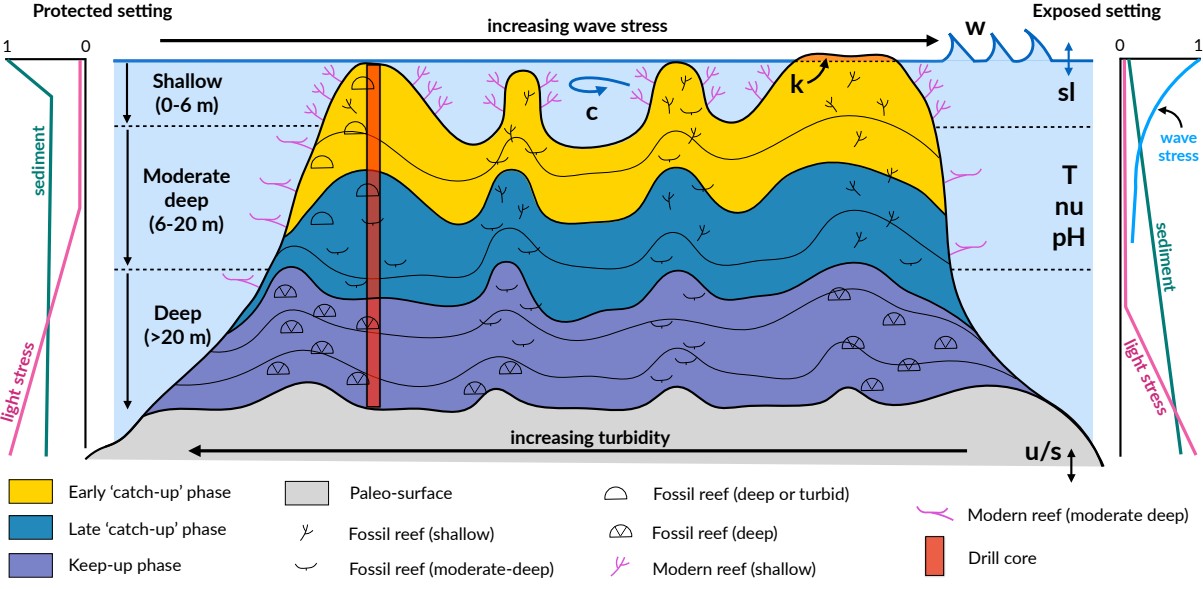

**Figure 1.** Schematic figure of a hypothetical reef with transitions from deep to shallow reef assemblages occurring up-core, illustrating a catch-up reef growth response to environmental forcing including light, sea-level changes (**sl**), hydrodynamic energy (**w** wave conditions and **c** currents), tectonic (**u** uplift and **s** subsidence), oceanic conditions (**T** temperature, **nu** nutrients, **pH** acidity), karstification (**k**) and sediment flux.

factors affecting growth have been classified by Veron (2011) as latitude-correlated factors, and those that are regional or local in character. Latitude-correlated factors include sea surface temperatures (SSTs), solar radiation and water chemistry (Kleypas et al., 1999). Regional and local environmental factors include wave climate, salinity, water clarity, nutrient influx, sedimentation regime and depth/composition of the initial substrate. These factors affect coral species to different extents,

5  controlling the distribution of coral communities across a reef (Hallock, 2001). Over longer time scales, they also shape the rate of calcium-carbonate production, framework building by corals, and the accumulation of sedimentary deposits (Done, 2011).

Despite the significant, short-term impacts cyclonic storms and terrigenous sediment input can have on reef systems (Cubasch et al., 2013), episodic disturbances are smoothed out on geologic scales (10,000's years) where reef systems are characterised

10  by remarkable persistence and resilience (Precht and Aronson, 2016). The persistent factors (e.g., sedimentation, wave climate and accommodation) are those that exert a stronger effect on the distribution of coral communities across a reef (Fig. 1). In the current study, we focus on these three main controls, however the model can simulate the impact of other ocean forcings (temperature, nutrients and acidity) on coral reef development.





## 3.1 Accommodation

Accommodation is the vertical and lateral space in the water column above the substrate within which corals can grow. The effect of accommodation on coral growth is the most well-understood constraint on the waxing and waning of reef development, and is governed by the relationship between the rate of vertical reef accretion, sea-level rise, subsidence and uplift (Woodroffe

and Webster, 2014).

Accommodation affects coral growth in two ways (Braithwaite, 2016). Firstly, light attenuates with depth in the ocean, and as corals are photosynthetic organisms, carbonate production decreases exponentially with increasing water depth (Schlager, 2005). Secondly, wave energy and water flow also decreases with depth, such that corals growing with reduced accommodation (i.e., in shallow depth) experience increased hydrodynamic energy (Montaggioni, 2005). The effect of light is assumed to

dominate over the effect of water movement in limiting carbonate production (Dullo, 2005) (Fig. 1), however both effects play a role in determining coral composition and, in turn, rates of vertical accretion (Cabioch et al., 1999; Kayanne et al., 2002). Generally, assemblages within 20 m depth have the highest accretion rates (10-20 m/kyr) than those deeper (< 10 m/kyr) (Montaggioni, 2005). For example in the Great Barrier Reef (GBR), Holocene reef growth largely occurred due to initially rapid sea-level rise ($\sim$10-6 ka), which created new accommodation and favourable conditions for reef 'turn-on' and the potential for

vertical aggradation (Hopley et al., 2007; Dechnik et al., 2015). Some reefs were able to keep pace with sea-level rise ('keep-up' reefs), while others caught up after sea-level stabilised ('catch-up' reefs), and others drowned ('give-up' reefs) (Davies et al., 1985; Neumann and Macintyre, 1985).

## 3.2 Hydrodynamic energy

At the organism level, currents, water flow and oscillatory motion induced by waves are critical in modulating physiological

processes in coral and thus influencing coral growth rates (Falter et al., 2004; Lowe and Falter, 2015). High water flow increases rates of photosynthesis by symbiotic algae (Bruno and Edmunds, 1998), nutrient uptake by corals (Weitzman et al., 2013), particle capture (Houlbrèque and Ferrier-Pagès, 2009) and facilitates sediment removal from coral surfaces (Rogers, 1990), all of which contribute to enhanced primary production. At the extremes, too little flow can be lethal in corals by inducing anaerobiosis, whereas extreme wave events cause mechanical destruction (Done, 2011) and can lead to long-term changes in

community diversity and structure (Madin and Connolly, 2006).

Waves exert a strong spatial control on hydrodynamics of reef systems (Lowe and Falter, 2015). Wave energy is largely dissipated on shallow reefs from bottom friction and wave breaking, with the former effect dominating the latter on reefs with high surface rugosity of coral communities (Rogers et al., 2016). Furthermore the geomorphology and high-rugosity of reefs cause wave refraction, such that wave energy is highest on the ocean-facing margin (Fig. 1 *exposed setting*) and lower in

back reef (Fig. 1 *protected setting*) lagoonal and marginal environments that are protected from the prevailing winds and wave energy (Harris et al., 2015, 2018). As a result, wave-induced bottom stress strongly influences coral cover and community composition.

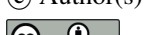


While overall, corals tend to grow more rapidly in higher-flow environments (Kuffner, 2001), high wave energy can also have a negatively effect on reef growth in shallow (<6 m) environments (Grossman and Fletcher, 2004; Rogers et al., 2016). Field studies demonstrate that coral communities form in particular hydrodynamic conditions and adopt specific morphologies suitable to those conditions (Done, 1982). Hence, wave-induced bottom stress affects community organisation spatially, with a
clear zonation pattern from the reef crest to the reef slopes (Done, 1982).

     While some studies have examined net water flow as it varies across whole reefs (Davies and Hopley, 1983) very few studies have examined the effect of water movement on corals themselves in a variety of environments (Sebens et al., 2003; Baldock et al., 2014) and even so, quantitative data on water energy thresholds for assemblages are non-existent. A high flow velocity threshold exists for some corals, beyond which they break (Baldock et al., 2014), however the reduction in coral growth in
response to water flow is poorly studied. The objective and benefit of *pyReef-Core* is its ability to model coral growth under changing flow conditions, thereby contributing to the understanding of the coral reef response to hydrodynamic energy.

### 3.3    Sediment input

High fluxes of both terrigenous and autochthonous sediments are widely identified to have both direct and indirect inhibitory effects on coral reef growth (Larcombe et al., 2001; Erftemeijer et al., 2012). Firstly, elevated turbidity attenuates ambient
photosynthetically active radiation (PAR), which inhibits the ability of corals to meet energy requirements through photosynthesis (Rogers, 1990). Secondly, smothering and abrasion by sediment blankets can impair feeding and cause physical damage and direct mortality (Sanders and Baron-Szabo, 2005). While the lethality of sediment exposure is determined by the intensity and duration to exposure, generally the long-lasting impact of turbidity regimes is known to depress coral growth and survival (Larcombe et al., 2001). For instance, elevated turbidity on mid-outer platform reefs caused by the suspension of sediment on
the Pleistocene reef substrate during initial flooding ∼9 ka is hypothesised to be responsible for a delayed initiation of coral growth in the southern GBR (Dechnik et al., 2015).

Autochthonous carbonate gravels and sediments (i.e. aragonite, calcite and high-magnesium calcite), produced by the growth and mechanical destruction of reef organisms through physical, biochemical and bio-erosive processes, are important determinants of the spatial and temporal distribution of coral communities on long timescales (Camoin et al., 2012; Kench, 2011).
Prevailing wave and current conditions of even moderate energy resuspend fine-grained carbonate sediments and are key in generating stable turbidity regimes on reef systems, particularly in lagoons, on leeward rims and on reef slopes at moderate depths due to the decreasing water energy gradient both laterally and with depth (Hopley et al., 2007). Similarly, prevailing turbid conditions are less common at shallow sites, especially on the windward rim due to wave-driven sediment removal (Fig. 1). The spatial variation of suspended sediment loads is a critical environmental factor influencing coral community distribution
across the reef and with depth (Perry and Larcombe, 2003). Turbid conditions are inimical to certain communities such as shallow-water corals, yet some species and communities are tolerant of elevated turbidity conditions on leeward rims (Dechnik et al., 2015) or species that thrive on reef slopes at depth (Perry et al., 2008). Hence, the spatial variation in turbidity is reflected in coral community distribution both across the reef and with depth.



Decades of experimentation carried out on the sensitivity of particular species to sediment have informed generic under-standing of the threshold levels of corals to the effect of natural sedimentation (Hubbard, 1986; Rogers, 1983; Stafford-Smith, 1993), however these thresholds have only been partially quantified in the literature, and tolerance at the assemblage level is difficult to constrain due to site and within-species variations (Erftemeijer et al., 2012). It has been shown that even under uni-

form sediment input regimes, inter and intra-site variations in sedimentation-resuspension regimes occur depending on water depth and exposure to wave energy (Wolanski et al., 2005). Early measurements supported that sedimentation rates exceeding 50 mg·cm$^2$/day produced lethal effects (Rogers, 1990). Yet each coral species has its own tolerance threshold to sediment stress, beyond which sedimentation produces sublethal to lethal effects (Erftemeijer et al., 2012). Hence, while there is a clear theoretical relationship between duration and rates of sedimentation and coral mortality and live coral cover, determining these

thresholds quantitatively have remained difficult to estimate.

## 4  *pyReef-Core* model

We present a 1-D deterministic, carbonate stratigraphic forward model called *pyReef-Core* that simulates vertical reef sequences comparable to those found in actual fossil reef drill cores. *pyReef-Core* is a tool to represent how dynamic biological and physical processes interact to create predictable, stratigraphic patterns. As shown in figure 2, the main steps in our workflow

are: (*i*) using real geological, geophysical and ecological data to establish environmental boundary conditions, vertical accretion rates of coral assemblages and defining assemblage tolerance thresholds to environmental factors; (*ii*) defining model input parameters including Malthusian and assemblage interaction matrix parameters, simulation time and those that define model resolution; before (*iii*) running the model to create a vertical core sequence that records assemblage changes and growth history.

### 4.1  Vertical reef accretion module

Carbonate production in a 1-D context, as represented by *pyReef-Core*, refers to the thickness of calcium carbonate produced in a core due to vertical framework accretion that is a result of vertical coral growth and sediment supply (Spencer, 2011). Hence, in this context carbonate production corresponds to reef vertical accretion. The model does not consider the destructional processes that occur on the reef due to physical, chemical and biological erosion but does account for erosional process during phases of subaerial exposure (referred as karstification in the model).

In our model, carbonate production is calculated for each time step at a user-defined resolution based on: (*i*) the maximum vertical accretion rate for each assemblage; (*ii*) GLVEs determining assemblage populations; and (*iii*) the environmental conditions that define optimal growth for each assemblage. During periods of subaerial exposure, karstification occurs at an uniform rate independent of the type of assemblages and consists in eroding reef stratigraphic top layers to the extent of the undergoing erosion.

In palaeoenvironmental analysis of real drill cores, assemblages are defined based on the relative abundance of coral species observable at certain intervals (Dechnik et al., 2015). To reflect this in our code, each depth interval in the modelled core records the assemblage that generated the greatest proportion of calcium carbonate at each time step; else if carbonate sedimentation



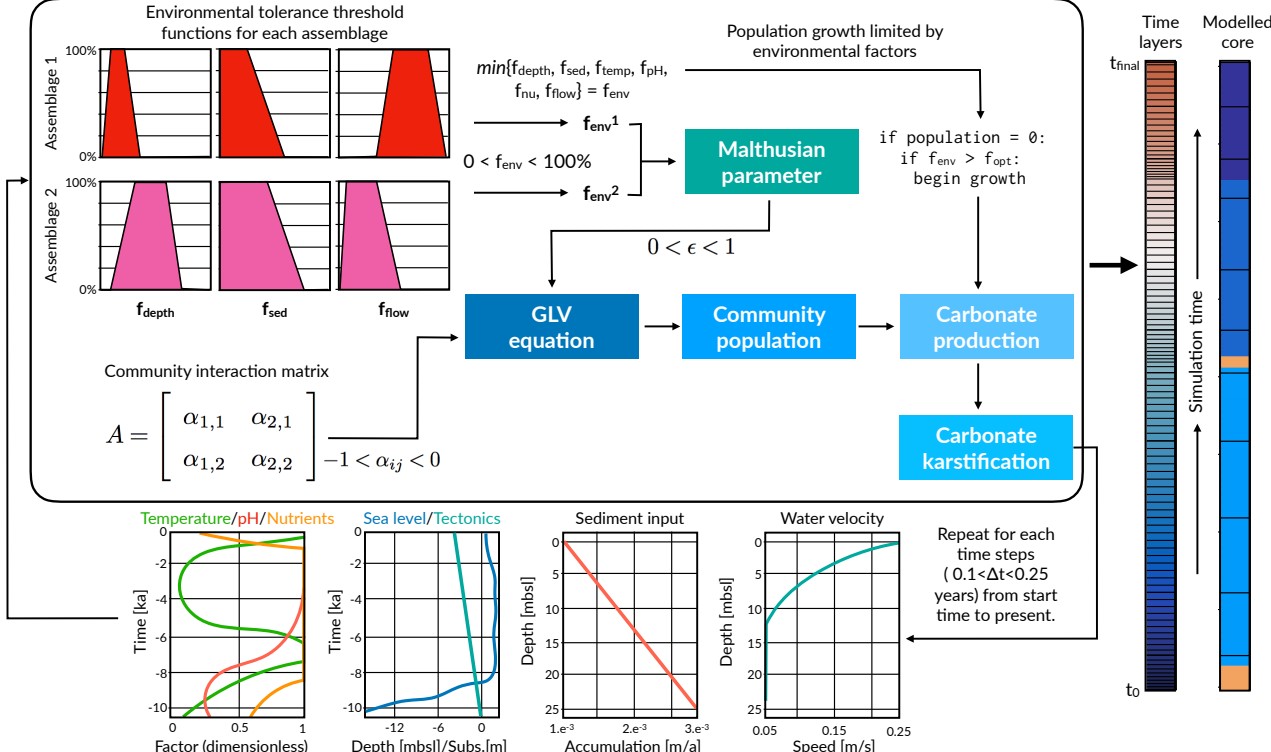

**Figure 2.** Illustration outlining *pyReef-Core* workflow (left) and of the resulting simulated core (right). First boundary conditions for sea-level, sediment input and flow velocity are set, which describes their relationship to either depth or time. The boundary conditions are used to establish the environment factor $f_{env}$ which describes the proportion of the maximum growth rate that an assemblage can achieve, depending on whether the environmental conditions exceed the optimal conditions for growth. The environment factor is scaled by the Malthusian parameter, which is in turn used as input in the GLVE equations to determine assemblage populations. Larger assemblage populations contribute to a faster rate of vertical accretion (here referred to as carbonate production). At the end of the time step, boundary conditions are updated and the process is repeated.

(defined as a depth-dependent sediment input function – Fig. 2) dominates coral production, sediment characterises the depth interval.

## 4.2   Generalised Lotka-Volterra equations (GLVEs)

The predator-prey ecological model by Lotka (1920) and Volterra (1926) is a well-known and simple model of species popula-

tion dynamics. Its generalised formulation (GLVEs) allows for an unlimited number of species and their pairwise interactions and is included here to simulate coral assemblage interaction dynamics. GLVEs applied to finding the evolution of species populations typically focus on ecologically-relevant periods (<5 years). The application of GLVEs for this problem is to simulate changes in coral assemblages observed in drill cores, where population dynamics are not the focus but only a means to estimate



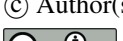

production rates over geologically significant periods. This is based on the understanding that internal ecosystem dynamics are partially responsible for the long-term biozonation patterns preserved in fossil reef records (Montaggioni, 2005).

Populations for each coral assemblage are determined by a logistic growth and decay function and a matrix of pair-wise assemblage interactions (Fig. 2), formalised in the equation:

$$\frac{dN_i}{dt} = \epsilon_i N_i + \sum_{j=1}^{c} \alpha_{ij} N_i N_j \qquad (1)$$

where $N_i$ is the population of coral assemblage $i$ for $c$ number of assemblages, $\epsilon_i$ is the intrinsic rate of increase/decrease of assemblage $i$ (also known as the Malthusian parameter) and $\alpha_{ij}$ represents the interaction coefficient among assemblages $i$ and $j$. Assemblage populations at time step $t_{i+1}$ are proportional to both populations at $t_i$ ($N_i$ and $N_j$) and to interaction coefficients ($\alpha_{ij}$) (Clavera-Gispert et al., 2017). The equation requires an initial population for each assemblage $N_i^0$, which is usually set to

zero for all populations as the basement substrate is unpopulated at the beginning of reef initiation simulations. Initialisation of any assemblage populations depends on environmental conditions and is related to the 'turn-on' criterion presented in section 4.7. Once these conditions are met for a particular assemblage, its population number is set to 1 and will evolve following the GLVEs defined above (Eq. 1).

### 4.3  Malthusian parameter ($\epsilon$)

Assemblage populations are proportionate to a Malthusian parameter $\epsilon$ which takes values between 0 and 1 and reflects the intrinsic reproduction of species through birth and mortality of corals in ecology (Fig. 2 – Clavera-Gispert et al. (2017)). However, in the geologic context of *pyReef-Core*, $\epsilon$ represents the tendency of corals to spatially dominate under favourable environmental conditions.

Clavera-Gispert et al. (2017) previously incorporated GLVEs to model the geological evolution of large scale carbonate plat-

forms and assumed that $\epsilon$ is not meaningful when timescales are beyond the lifespan of an organism and supposed that $\epsilon = 1$. Clavera-Gispert et al. (2017) examine carbonate production in 500-year intervals whereas *pyReef-Core* explores much smaller intervals (<10 years), and as coral colonies may live for several decades to centuries (Camoin et al., 1997; Grigg, 2002), $\epsilon$ is not assumed to be 1. Even when assuming $\epsilon = 1$, *pyReef-Core* simulations produced volatile population dynamics where assemblage populations grew exponentially and were unable to replicate long-term ecosystem stability, nor the thousands-of-years

of assemblage persistence observed on some reefs (Camoin et al., 1997). Hence, while values of $\epsilon$ are not yet known at the decadal scale, $\epsilon$ is an important parameter regarding spatial changes in assemblage distributions that occur within centuries. Finally, $\epsilon$ is scaled according to the environmental factors to take into account the limiting effect of inimical environmental forces on assemblage population growth (section 4.6).

### 4.4  Assemblage interaction matrix

The pair-wise coefficients of interaction between assemblages can be represented as elements $\alpha_{ij}$ in a square C-by-C matrix, where any $\alpha_{ij}$ is a special case of the effect of a change in assemblage population $N_i$ on itself. Values of the coefficients





**Table 1.** Interaction possibilities among coral assemblages and the associated range of matrix coefficients, adapted from Clavera-Gispert et al. (2017).

| Interactions | Effect on $i$ | $\alpha_{ij}$ range | Effect on $j$ | $\alpha_{ij}$ range |
|---|---|---|---|---|
| **Competition** | Detrimental | $-1 \leq \alpha_{ij} \leq 0$ | Detrimental | $-1 \leq \alpha_{ij} \leq 0$ |
| **Neutralism** | Neutral | $\alpha_{ij} = 0$ | Neutral | $\alpha_{ij} = 0$ |

describe the beneficial, neutral or detrimental effects of one species on another (Table 1). As with $\epsilon$, values of $\alpha_{ij}$ cannot be inferred from previous ecological modelling studies (e.g., Clavera-Gispert et al., 2017) as the temporal scales of study are irreconcilable. It is assumed, however, that competitive-to-neutral effects control the spatial distribution and abundance of coral assemblages at decadal timescales.

Competitive interactions between corals have received considerable attention in ecology (Lang and Chornesky, 1990), especially regarding their spatial distribution outcomes. As assemblages occupy ecological niches, each will spatially dominate at a location under specific environmental conditions, outcompeting other assemblages for food, space and light (Connell et al., 2004). Hence, competition is an important determinant of reef biozonation which persists over centennial timescales given that coral growth is slow and colony lifespans can be centuries long. Hence in *pyReef-Core*, the interaction matrix is formed by
competitive-to-neutral interaction coefficients between -1 and 0 (Table 1).

### 4.5 Computing carbonate production based on assemblage populations

Solved GLVEs determine population growth/decline for each assemblage, and are used to compute carbonate production [cm/y] for each time step. The amount of carbonate produced by each coral assemblage during each time step is defined as:

$$\frac{dp_i}{dt} = \frac{\sum_{i=1}^{c} A_i \times N_i}{S} \tag{2}$$

where the carbonate production at every time step of each assemblage $p_i$ for $c$ number of assemblages is a product of the population distribution $N_i$ and the maximum rate of vertical accretion $A_i$ in proportion to a scalar $S$. The scalar is introduced to the vertical growth equation in order to minimise distortionary effects of exponential growth trends for each population occurring in the absence of inter-assemblage competition (i.e., to prevent unreasonably large population growth when only one assemblage can exist under certain conditions). Total vertical reef growth $G$ recorded in a core is the sum of carbonate sediment
deposited $p_{sed}$ and all calcium carbonate produced by each assemblage:

$$\frac{dG}{dt} = \sum_{i=1}^{c} p_i + p_{sed} \tag{3}$$

### 4.6 Environmental factors

Sediment input, water flow and accommodation are the basic environmental factors influencing coral growth in *pyReef-Core*. However, the model architecture is such that in the future it is possible to simulate the effect of other important environmental



parameters such as ocean temperature, acidity and nutrient. Tolerance functions are defined for each environmental factor as a set of four points that indicates both the range in which an assemblage would reasonably exist based on published empirical data (Done, 1982; Hopley et al., 2007; Dechnik, 2016) and the rate at which vertical accretion reduces as the environmental conditions exceed upper or lower threshold limits for each assemblage (Fig. 2). As such, they define an 'optimal growth

window' for each assemblage. The threshold functions for each assemblage to ambient environmental conditions are combined into a single environmental parameter $f_{env}$ subject to the minimum value rule:

$$f_{env} = \min \left[ f^i_{depth}, f^i_{sed}, f^i_{temp}, f^i_{pH}, f^i_{nu}, f^i_{flow} \right] \quad (4)$$

where $f_{depth}$, $f_{sed}$, ... and $f_{flow}$ represent the threshold functions for each assemblage $i$. Hence, $f_{env}$ is seen as the combined effect of ambient environmental conditions on optimal growth conditions (Fig. 2). Finally, the Malthusian parameter $\epsilon$ is scaled

by the environmental factor such that:

$$E^i = \epsilon \times f^i_{env} \quad (5)$$

which reflects the limiting effect on environmental factors on the growth potential of each assemblage.

### 4.7 'Turn on' criterion

At the initialisation of the *pyReef-Core* simulations, assemblage populations are usually set to zero. Population growth only

occurs when the initial criterion $f_{env} > f_{opt}$ is met (Fig. 2). It reflects the notion that reef 'turn on' events occur because of a confluence of optimal conditions including a shallow substrate, favourable energy, light and water temperature, pH and nutrients conditions and relatively low sediment supply (Buddemeier and Hopley, 1988; Fabricius, 2005; Dechnik et al., 2015). In other words, *pyReef-Core* only initiates growth when a degree of optimality in growth conditions are met. By default, the value of $f_{opt}$ is set to $0.5$ which stands that the 'turn-on' criterion is met when environmental conditions enable at least 50% of

the maximum vertical accretion.

### 5   Examples of model application

Two case studies are presented here to assess the ability of *pyReef-Core* to reproduce realistic sequences found in drill core. We simulate the interactions between three assemblages which are estimated based on water depth intervals (shallow, intermediate and deep). We also consider that coral production in these experiments is primarily controlled by accommodation and exposure

to sedimentation (Chappell, 1980; Tudhope, 1989) and water flow (Fulton et al., 2005; Comeau et al., 2014).

### 5.1   Experimental settings for model simulations

#### 5.1.1   Assemblage maximum vertical accretion rates

Maximum vertical accretion rates in the simulation are user-defined. For shallow assemblages on exposed margins, maximum vertical accretion rates (11 m/kyr) are chosen to reflect known average rates for robust branching coral facies in high-energy





environments established for the Indo-Pacific (Montaggioni, 2005). Moderate-deep assemblages represent slightly higher maximum accretion rates (15 m/kyr) with the lowest accretion rates (9 m/kyr) for deep assemblages. These were chosen to reflect the average accretion rates for Indo-Pacific tabular-branching and massive coral facies found in high-energy conditions respectively (Montaggioni, 2005).

### 5.1.2 Ecological dynamics

*pyReef-Core* requires knowledge of the intrinsic rate of assemblage population growth/decline ($\epsilon_i$) and the matrix coefficients ($\alpha_{ij}$) of interactions between distinct assemblages. However, inferring ecological dynamics from ecological studies is challenging. Empirical studies of coral competition and growth are often focused at the species, rather than assemblage, level and explain competitive relationships qualitatively rather than quantitatively (e.g., Connell et al., 2004). Moreover, GLV equations have not been used to model coral population dynamics at the temporal resolution (centennial to millennial) we are interested in. Based on an initial sensitivity analysis, we define a set of values for the Malthusian parameter ($\epsilon_i$) and interaction coefficients among assemblages ($\alpha_{ij}$) which are summarised in Table 2. Chosen coefficients define small competitive interactions between assemblages.

The coral assemblages defined in this study largely do not share the same environmental setting and optimal growth conditions. Therefore, competitive interactions are restricted to only those assemblages that may reasonably co-exist due to overlapping depth, sediment flux or flow velocity thresholds. This translates to an interaction matrix with values only along the main diagonal, super- and sub-diagonals. Everywhere else, interactions are set to 0. Associated to these interactions, we define a series of critical threshold response functions for each assemblages (Fig. 3).

### 5.1.3 Depth threshold functions

Based on a statistical analysis of the depth and environmental distribution of modern coral communities at One Tree Reef (Great Barrier Reef, GBR), Dechnik et al. (2017) calibrated the palaeo-water depositional environments of six fossil coral assemblages (three in protected and three in exposed environments). This calibration was also based on quantitative measurements of crustose coralline algae thickness and vermetid gastropod abundance which are reliable palaeo-depth indicators, allowing for the depth intervals to be more accurately constrained. These assemblages are broadly consistent with shallow- and deep-water coral facies of the Indo-Pacific (Cabioch et al., 1999; Camoin et al., 2012).

Here, three assemblages typically occurring on exposed slopes are modelled according to the estimated depth intervals defined by Dechnik (2016) (Table 2) and represent shallow-water ($< 6$ m), moderate-to-deep-water ($6 - 20$ m) and deep-water ($20 - 30$ m) assemblages respectively.

### 5.1.4 Water flow

The water flow function is constructed according to the theoretical relationship defined by Chappell (1980) whereby wave stress decreases exponentially with depth (Fig. 4). Here, we rely on the velocity-depth relationships on wave-exposed reef



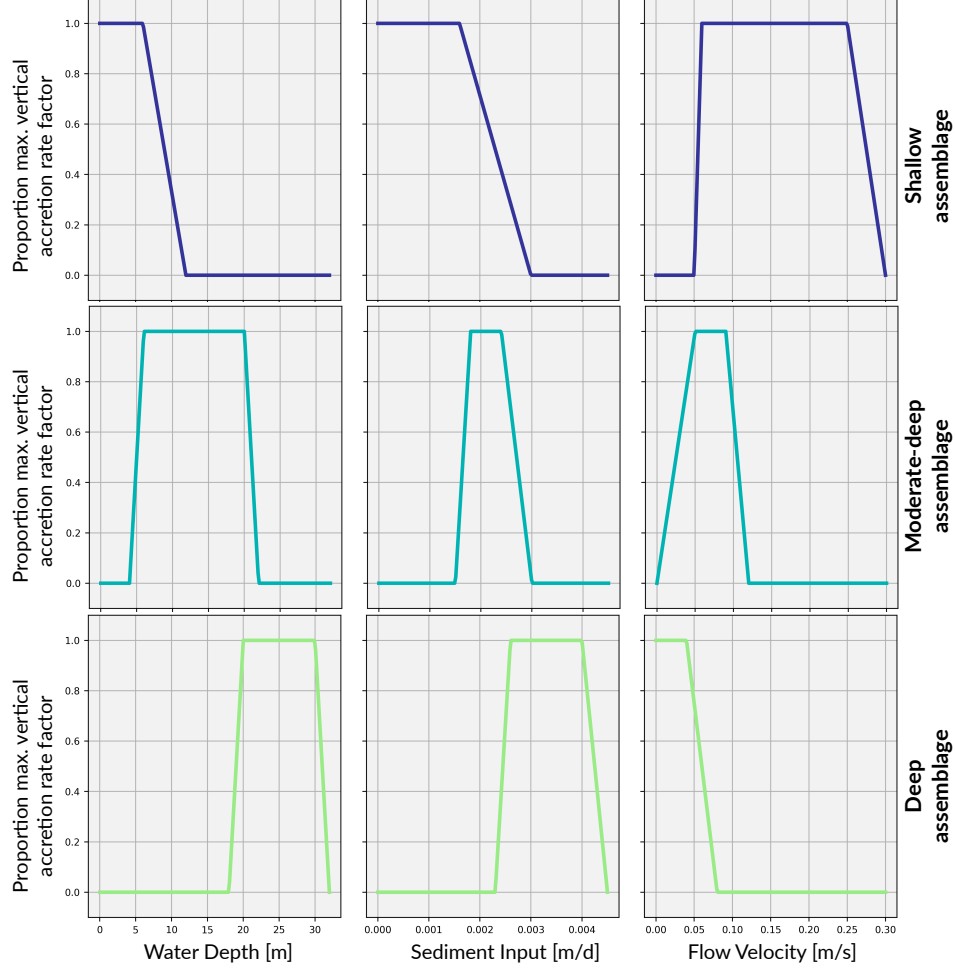

**Figure 3.** Environmental threshold functions for shallow, moderate-deep and deep assemblages characteristic of a synthetic exposed margin. The x-axis indicates the limitation on maximum vertical accretion for conditions outside the optimal maximum vertical accretion rate.

slopes from the field study by Sebens et al. (2003). A maximum velocity of 25 cm/s in region $\leq 1$ m and an exponential decrease up to 25 m below which flow velocity is set to 0. This is consistent with direct observations from exposed algal flat (Davies and Hopley, 1983), and maximum velocities (> 50 cm/s) beyond which branching corals are susceptible to breakage (Baldock et al., 2014).

5   With lacking specific data on the optimal flow environment for specific corals, assumptions about thresholds for distinct coral assemblages are inferred from boundary conditions. That is, the water flow exposure threshold range for each assemblage reflects the attenuation of water flow with depth (Fig. 3).





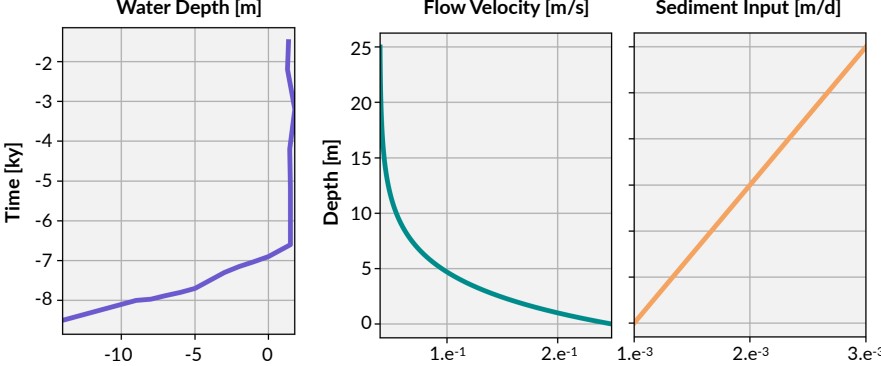

**Figure 4.** Left curve shows the Holocene sea-level curve estimated from Sloss et al. (2007). Right graphs illustrate the boundary conditions established for flow velocity and sediment input used in the experimental simulations.

### 5.1.5 Sediment exposure

*pyReef-Core* can model the vertical sedimentation rate (m/day) as a function of either time or depth. When sediment flux is dependent on depth, it implies that sediments are autochthonous (loose carbonate materials), in contrast to terrigenous sediments transported from outside the reef system (siliclastic materials), which may be represented by sediment flux varying

5 with time. In our case studies, we use a depth-dependent sedimentation rate input curve to approximate the temporal variations in sediment accumulation along the core (Fig. 4).

Sediment tolerance thresholds for each coral assemblage (Fig. 3) are informed by (Dechnik et al., 2017) before receiving maximum and minimum sedimentation rates corresponding to the sediment input boundary condition (Fig. 4. The boundary condition provides a broad indicator of the sediment load expected at certain depths, and thus what would be tolerated for each

10 depth-specific assemblage. With alternate sediment input boundary conditions, the upper and lower tolerance thresholds can be adjusted to represent how coral communities respond differently to site-specific suspended sediment levels.

### 5.2 Case 1: GBR idealised windward shallowing-upward Holocene reef sequence

Based on afore described experimental settings, we first simulate a typical shallowing-up sequence of coral assemblages on the exposed rims of several reef in the GBR, expressing a catch-up strategy of reef growth during Holocene sea-level rise (∼9.4

15 ka to present).

### 5.2.1 Initial parameters

Considering the simulated temporal scale, neither subsidence nor uplift are considered to be important (Hopley et al., 2007) in this experiment. Instead, accommodation is simulated as a function of Holocene sea-level changes and vertical coral reef growth only. The Holocene relative sea-level (RSL) curve from Sloss et al. (2007) is used to represent sea-level change (Fig. 4).



**Table 2.** Parameter values used in our two experiments. Estimates of maximum production rates for assemblages were determined based on literature surveys of maximum growth rates for coral facies of GBR (Davies and Hopley, 1983) and Indo-Pacific reefs (Montaggioni, 2005).

| Parameters | values |
|---|---|
| **Malthusian parameter** | $\epsilon_i = 0.004$ |
| **Assemblage interaction matrix** | |
| Main diagonal | Detrimental - $\alpha_{ii} = -0.0005$ |
| Sub- and super-diagonal | Detrimental - $\alpha_{ij} = -0.0001$ |
| **Assemblage maximum growth rate** (m/yr) | |
| *Shallow-water assemblage (0-6 m)* | 0.011 |
| *Moderate-deep-water assemblage (6-20 m)* | 0.012 |
| *Deep-water assemblage (20-30 m)* | 0.009 |
| **Assemblage threshold tolerance variables** | |
| *Shallow-water assemblage (0-6 m)* | |
| Absolute water flow threshold range | $0.05 \leq f_{flow} \leq 0.3$ |
| Absolute sediment input threshold range | $0 \leq f_{sed} \leq 0.003$ |
| *Moderate-deep-water assemblage (6-20 m)* | |
| Absolute water flow threshold range | $0 \leq f_{flow} \leq 0.12$ |
| Absolute sediment input threshold range | $0.0015 \leq f_{sed} \leq 0.003$ |
| *Deep-water assemblage (20-30 m)* | |
| Absolute water flow threshold range | $0 \leq f_{flow} \leq 0.08$ |
| Absolute sediment input threshold range | $0.0023 \leq f_{sed} \leq 0.0045$ |

The data implies a RSL history that is characterised by a mid-Holocene highstand of 1.8 m at ~4 ka before returning slowly to present sea-level, matching other estimates of RSL (Chappell, 1983; Lewis et al., 2013).

Simulation begins at 8.5 ka, which is within the take-off envelope for Holocene growth of outer-platform GBR reefs (Hopley et al., 2007). At 8.5 ka, RSL is 15 m below sea-level (Sloss et al., 2007) and substrate is at 20 m depth in order to simulate

5 a catch-up growth strategy from a deep substrate. We compute the GLVEs at time intervals of 2.5 years and combine each accumulated assemblage as a stratigraphic unit within the core for every 50 years.

### 5.2.2 Communities evolution and synthetic core representation

Figure 5 presents the GBR-representative assemblages summarised by Dechnik (2016) as well as the simulated core by *pyReef-Core*. The modelled core is 35 m long and is composed of three assemblages characteristic of an exposed margin and carbonate

10 sediments. The simulation portrays two distinct assemblage transitions from massive assemblages representing deep (20-30), low-flow conditions to a faster-growing, tabular-and-branching assemblage characteristic of the 6-20 m depth interval, which is succeeded in shallow water ( $< 6$ m) by a robust-branching assemblage representing higher-energy conditions (Figs. 6, 5).





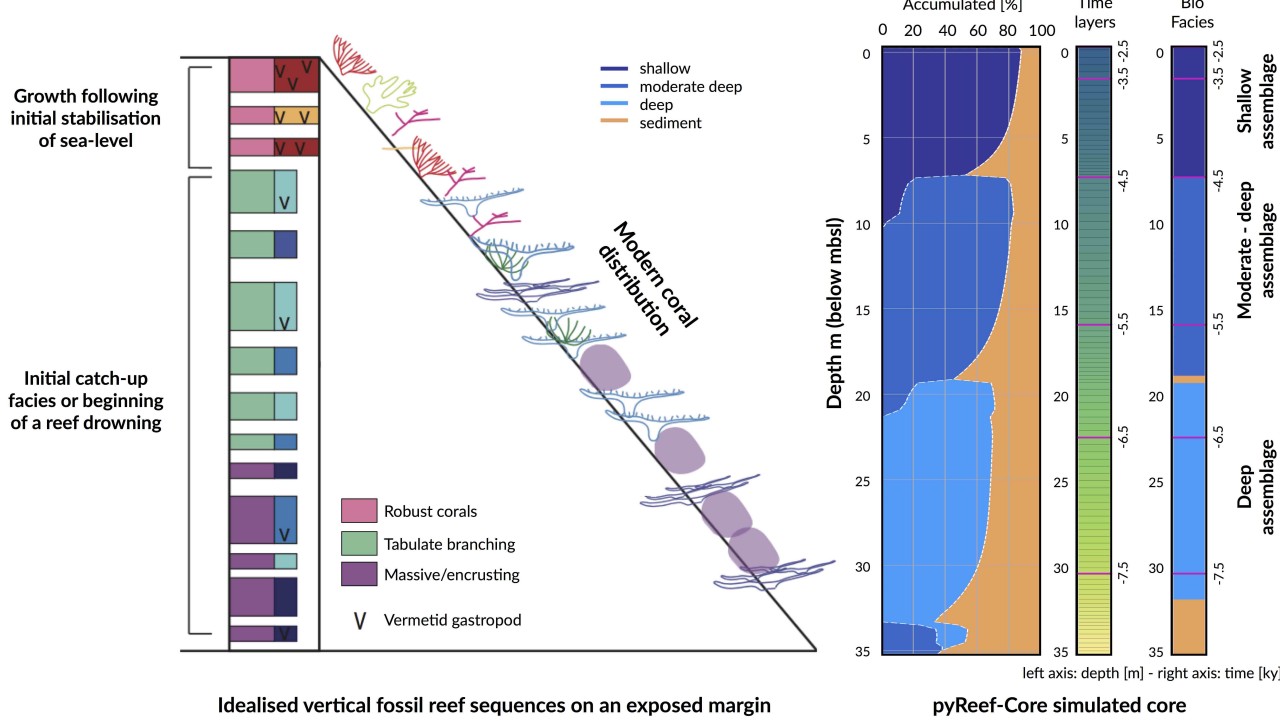

**Figure 5.** (Left) Ideal shallowing-up fossil reef sequence representing a 'catch-up' growth strategy with associated assemblage compositions and changes, adapted from Dechnik (2016); (Right) Model output of produced *pyReef-Core* sequence representing a similar shallowing-upward, 'catch-up' phase.

As sea-level rises from 8.5 to 6.5 ka, the deeper assemblages have sufficient accommodation space (>20 m) and low-flow to thrive. However, greater sediment input at depth is inhibitive in the early part of the simulation at the base of the core (32-35 m) (Fig. 5. As sea-level begins to stabilise (top right panel Fig. 6), accommodation space decreases and moderate-deep assemblages start to dominate the sequence up to 4.7 ka (bottom left panel Fig. 6). Following stabilisation from 4.7 to 3.2 ka, shallow assemblages develop as a result of the decreased accommodation space ($\sim$ 6 m at 4.7 ka), high-velocity hydrodynamic conditions and reduced sediment input. Assemblage growth rates (bottom right panel Fig. 6) show a pattern similar to the population number curves with values lower than assemblage maximum production rates (Table 2) indicative of the effects of environmental factors (sediment input and flow velocity) on the growth of each assemblage. The deeper assemblage is 15 m thick and is composed of 30-60 % loose sediment and is succeeded by $\sim$ 12 m of moderate-deep assemblages with a lesser proportion of sediment (Fig.5). The last 6-7 m of core are predominantly formed by shallow assemblages with on average less than 20% of carbonate sediments (Fig. 5). The simulated shallowing-up sequence accurately reflects expected shift from deep to moderately-deep assemblages at $\sim$15-20 m depth, and from moderately-deep to shallow assemblages at $\sim$6 m depth proposed by Cabioch et al. (1999) and Dechnik (2016). The simulated sequence relates well with the description proposed





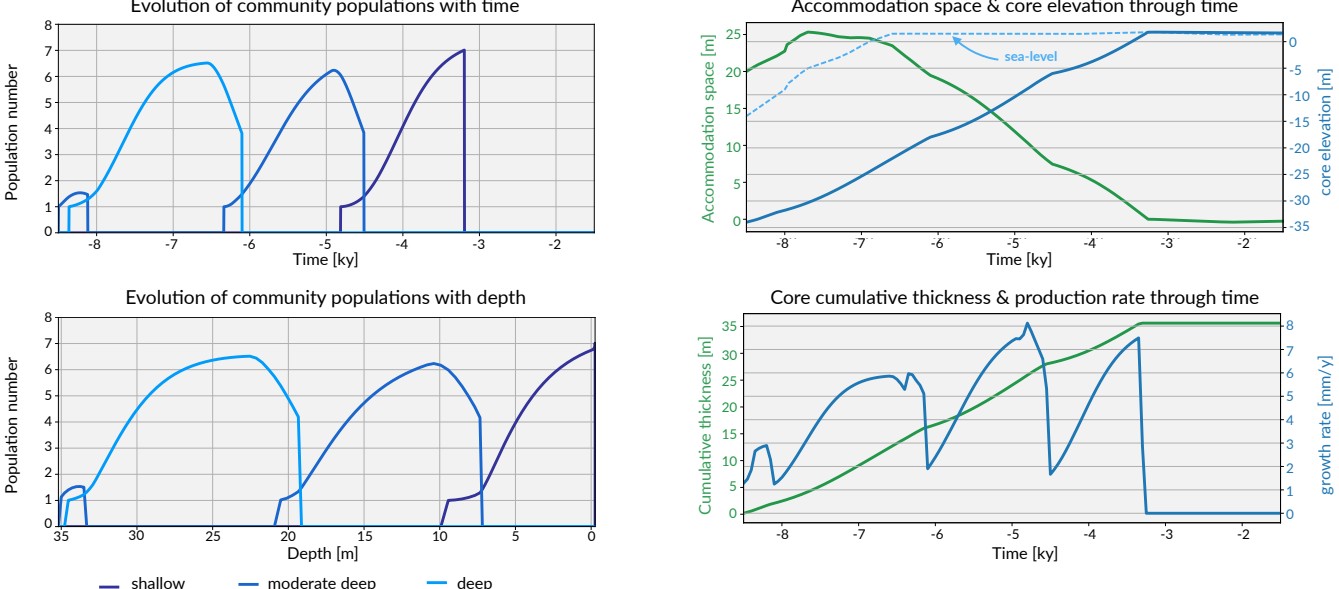

**Figure 6.** Graphical output from *pyReef-Core* showing on the left the evolution of each community in the form of population number with time and depth. As mentioned previously population number here is a proxy for carbonate production with larger assemblage population corresponding to faster rate of vertical accretion. Right top panel shows the evolution of the accommodation space and core elevation through time in relation to imposed sea-level curve. Right bottom panel presents the temporal evolution of the cumulative thickness as well as the total coral production rate for the considered experiment.

by Dechnik (2016) and reproduces the distinct assemblages defined in the idealised reef sequences found on exposed margin along the GBR (Fig. 5).

The modelled core reaches sea-level at around 2.5 ka (Fig. 6 which also correlates well with values reported for several reefs in the GBR (Davies and Hopley, 1983; Dechnik et al., 2015; Salas-Saavedra et al., 2018). Average vertical accretion rate implied

5    by the model is around 4.1 m/kyr (Fig. 6), again in the range of actual drill cores average rates which varies around 3 to 5 m/kyr on exposed reef margins (Davies and Hopley, 1983; Camoin et al., 2012; Dechnik et al., 2015). It is also worth noting that coral growth becomes predominant within the sequence at ~7.8 ka in the modelled core which coheres with the observed delay in reef initiation of approximately 1 kyr (Dechnik et al., 2015) after initial flooding of the substrate during the Holocene transgression. We also notice that the transitions between assemblages also correspond to periods where the proportion of carbonate sediment

10   deposited increases (Fig. 5). It mimics a lag between optimal conditions from one assemblage to the other and relates to the choice of environmental threshold functions that were imposed in our simulation (Fig. 3). Overall, the model reproduces the details of the formation of shallowing-upward sequences both in terms of assemblages succession, accretion rates, deposited thicknesses and timing of initiation. It can be applied to estimate the impact of changing environmental conditions on growth rates and patterns under many different settings and initial conditions.





### 5.3 Case 2: GBR idealised reef core reconstruction over the last 140 kyr

For the second study case, the experimental settings for threshold functions, ecological dynamics, water flow and sediment exposure (presented in section 5.1) remain unchanged. The goal is not to match a specific drill core but to illustrate the influence of forcing conditions on the development of a coral reef sequence with our model.

#### 5.3.1 Initial parameters

We reconstruct using *pyReef-Core* the evolution of an ideal coral reef sequence since the last interglacial (LIG). LIG is represented by marine isotope stage (MIS) 5e, which is a proxy record of low global ice volume and high sea-level (Grant et al., 2012). It is arbitrarily set to begin at approximately 130 ka before present and our simulation runs over 140 kyr. The GLVEs which control the coral productions dynamic, are updated every 25 years and stratigraphic layers are recorded at time interval of 100 years.

Here we use the sea-level curve proposed by Grant et al. (2012) who estimate sea-level records based on the timing of past ice-volume changes, relative to polar climate change. The relative sea-level change over the simulated period has rates of rise reaching 12 cm/y during all major phases of ice-volume reduction, with values below 7 mm/y when sea-level exceeded present mean sea level (Grant et al., 2012). The applied sea-level curve is shown in top right panel of Fig. 7.

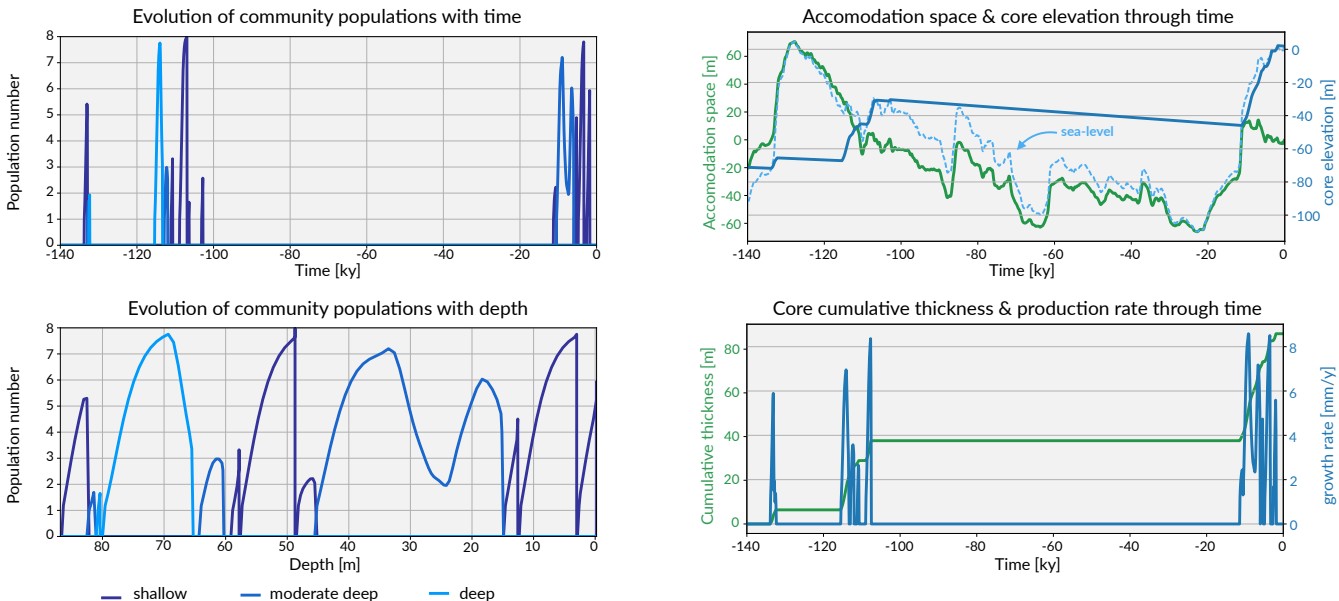

**Figure 7.** Similar to the previous case, these graphs shows on the left the evolution of each community in the form of population number with time and depth. Right top panel shows the evolution of the accommodation space and core elevation through time in relation to imposed sea-level curve. Right bottom panel presents the temporal evolution of the cumulative thickness as well as the total coral production rate.





Karstification of Pleistocene reef limestone has been identified as a controlling factor on variations of antecedent topography, which in turn is thought to influence the morphology of modern reefs (Purdy and Winterer, 2001). Rates of karstification are a function of exposure time, rainfall, porosity and original topography of exposed carbonate reefs. Summary of karstification rates from both the Indo-Pacific and Caribbean shows values ranging from 0.01 m/kyr (Barbados, Hopley et al. (2007)) to 0.14

m/kyr (mid-outer platform reefs, Southern GBR, Marshall and Davies (1982)). Here we impose a karstification rate of 0.07 m/kyr consistent with estimates from Ribbon Reef 5 and outer Central GBR shelf (Webster, 1999).

Over such period of time, sea-level fluctuations are not the only factor controlling the accommodation change and uplift/subsidence evolution has to be considered (Spencer, 2011). Based on a comprehensive study of GBR reefs, Dechnik (*unpublished*) estimates that a subsidence rate of ∼0.083 to 0.13 m/kyr is required to explain the observed elevation of the upper surface of the

LIG reef that provides the antecedent topography of the modern mid-outer platform reefs in the GBR. The proposed range is consistent with values found for other reefs along the GBR (Marshall and Davies, 1982; Webster, 1999). In our model, we use a constant rate of subsidence set to 0.1 m/kyr that corresponds to 14 m of subsidence over the duration of the simulation. In addition, the initial elevation is set 20 m above sea-level position at the start of the simulation (140 ka), corresponding to a depth of ∼70 m below current sea-level position.

**5.3.2    Communities evolution and synthetic core representation**

Prior to 135 ka, the model shows a first stage of reef growth characterised by shallow water, high energy coral communities colonisation (left panels Fig. 7 and Fig. 8), following flooding of antecedent platform. The cumulative thickness for this phase is < 10 m (bottom right panel Fig. 7) and is compatible with values estimated for Ribbon Reef 5 and Heron Island (Dechnik et al., 2017).

Following this initial phase, a deepening upwards sequence occurs up to 132 ka (Fig. 8). Again this sequence has also been identified in similar time interval at One Tree Reef (southern GBR) and Stanley Reef (central GBR) (Dechnik et al., 2017). A lack of significant reef framework (< 30%) characterises the stratigraphic sequence during this interval.

The rapid sea-level rise (Grant et al., 2012) during the end of the penultimate deglaciation explains the drowning event observed in the core from 128 to 118 ka (bottom left panel in Fig. 7). During this period, the accommodation increase is mainly driven

by sea-level fluctuations and to a small extent (∼1 m) by the imposed subsidence rate.

From 118 to 107 ka, during the first stage of the regression phase, a shallowing upward sequence (∼ 30 m thick) is identified with three distinct community populations modelled over time (bottom left panel Fig. 7). During this time interval, the maximum population number for the moderate-deep assemblages is relative lower (<3) than for the 2 other assemblages (>7). Consequently, the percentage of accumulated thickness for this assemblage is below 7%. These assemblage transitions are

primarily controlled by high frequency sea-level variations observed in Grant et al. (2012) curve (top right panel Fig. 7). Minor events of karstification (<2 cm of erosion) are triggered by short episodes of sub aerial exposure around 110 ka. From 107.5 to 104 ka, high energy coral communities (shallow assemblages) dominate the sequence with a maximum growth rate above 8 mm/yr (bottom right panel Fig. 7).



**Figure 8.** Simulated reef core reconstruction, showing the different stages of last interglacial reef growth in relation to sea-level, karstification and subsidence.



The following stage from 107 to 12 ka is characterised by a period of sub aerial exposure due to sea-level fall (bottom right panel Fig. 7). Both subsidence and karstification occur and account for nearly 11 m of elevation offset with about 1 m attributed to karstification processes (Fig. 8). Applied to a real case, *pyReef-Core* can be used to test several scenarios with different rates of subsidence and karstification in order to explain for example the discrepancy in age/elevation data of LIG deposits observed in

the GBR (Marshall and Davies, 1982; Dechnik et al., 2017). It can also be used to estimate the contribution of karst dissolution and subsidence (Hopley et al., 2007; Purdy and Winterer, 2001) with a more quantitive approach.

By 13 ka, sea-level re-floods the LIG reef and Holocene reef growth initiates ∼10.5 ka after in the experiment (Fig. 7). The lag (2.5 kyr) between flooding and reef growth initiation matches well with observations for the GBR (Fabricius, 2005; Hopley et al., 2007; Camoin et al., 2012). However the timing for the initial flooding occurs 3 kyr earlier than what is expected for the

GBR. This temporal difference is related to both sea-level variations (Grant et al., 2012) and chosen initial starting elevation of the model. The Holocene reef sequence is around 46 m thick (Fig. 8) which is above most of the GBR reef maximum vertical accretion thicknesses (usually < 30 m) but correlates with thicknesses found in reefs from Tahiti and Huon Peninsula (Woodroffe and Webster, 2014). This Holocene sequence is first composed of more than 30 m of moderate-deep assemblage which corresponds to the catch-up phase discussed in the first study case and is associated to the rapid sea-level rise. The

reef accretion rate during this time interval is maximum and reaches values above 8.2 mm/y (bottom right panel Fig. 7). The remaining ∼15 m of the upper most sequence is built of shallow assemblages that become predominant after 6 ka when sea-level rise decreases. It is also worth noting the presence of short periods of subaerial exposure which coincide with two small karstification events (karst dissolution < 1 cm, Fig. 8).

The total simulated core has an overall thickness > 86 m. A complete sequence such as the one modelled here is unlikely to

be found in natural reef complex mainly due to the 3D nature of such system (Woodroffe and Webster, 2014). Nevertheless the predicted sequence represents in 1D, the idealised succession of coral assemblages produced for a given set of initial and forcing conditions. Therefore it can be compared to series of drill cores at different positions along a given region and used as a quantitative approach to analyse stratigraphic responses of coral reefs to a combination of physical, biological and sedimentological processes.

## 6  Discussion

Relatively little is known about how coral reefs grow and respond to environmental conditions at temporal scales exceeding what is measurable (i.e. observational record over the last 100 years) (Hughes et al., 2017). It has been a major challenge for both geological and ecological studies to adequately capture coral reef ecological and environmental dynamics on centennial to millennial temporal scales and at reef scales (Stocker et al., 2013). Our new method, *pyReef-Core*, operates on these scales,

and offers a coherent, fast and effective way to predict 1-D reef core stratigraphies and assemblages changes. It can be used to improve our understanding of coral reefs response to climatic and environmental changes (Done, 2011; Harris et al., 2015). The code is most useful in application to reef researchers examining the vertical distribution of coral assemblages and coral growth dynamics (Montaggioni, 2005; Camoin et al., 2012; Dechnik, 2016) by comparing outputs between modelling cores.





This would enable the extrapolation of knowledge gained from examining drill cores to areas of the reef where data is scarce. It can also be used to understand environmental histories of cores where dating or classification of assemblages is difficult due to poor core recovery. Despite its 1-D limitation, the model can be applied to gain a 3-D picture of the environmental, ecological and geomorphological history of a specific reef. This can be achieved by defining multiple biological and environmental initial

conditions representing, for example, the differences in assemblage types and hydrodynamic conditions between the windward and leeward margins of the reef (Cabioch et al., 1999; Dechnik et al., 2015; Salas-Saavedra et al., 2018).

Necessarily, *pyReef-Core* is also a simplified representation of a coral reef system and required a number of free parameters such as sediments, flow, Malthusian parameter, and community matrix parameters which needs to be defined for modelling. The task of finding these set of parameters that best describes a specific reef site and core data is challenging for several reasons. Firstly,

empirical estimates of environmental tolerance thresholds of given assemblages are scarce in the scientific literature making their estimation difficult (Camoin et al., 2012; Baldock et al., 2014; Dechnik et al., 2017). Therefore results interpreted from the modelled environmental threshold represent hypotheses that must be tested and validated against additional real, physical measurements on reefs. Secondly, reefs experience a variety of natural sedimentation regimes due to the variable morphologies (Hopley et al., 2007) and flow regimes due to the position of reefs in respect to the dominant swell (Dechnik, 2016) and

proximity to the coast (Larcombe et al., 2001). Consequently, it is difficult to construct a model that fully represents complex reef system dynamics simultaneously. Thirdly, estimation of the interaction matrix coefficients and Malthusian parameters remains difficult (Clavera-Gispert et al., 2017), specifically when considering coral assemblages dynamics at the temporal scale (decadal to centennial) relevant to *pyReef-Core*. Yet, interpretations of these parameters from ecological modelling studies provide a useful guide in regards to reef biozonation and assemblages competition (Lang and Chornesky, 1990; Grigg, 2002;

Connell et al., 2004). Finally, modelled vertical accretion or growth patterns in *pyReef-Core* are non-linear reflecting the natural complexity of coral reef systems and the biological and physical interactions occuring at reef scales. It poses the problems for calibrations and the underlying uncertainties inherent to our simplified approach (Burgess and Wright, 2003; Warrlich et al., 2008; Clavera-Gispert et al., 2017). Nevertheless, our model represents a shift from the standard accommodation-forced geometrical models (Dalmasso et al., 2001; Gale et al., 2002; Burgess et al., 2006) where coral reef stratigraphy is controlled

mainly by changes in sea-level. Even if our approach is a simplification of natural processes, the simulated stratigraphic patterns are a sum of simultaneous, interacting tectonic, biological, physical and sedimentological processes.

*pyReef-Core* can be described as a multi-dimensional (i.e. many parameters) and multi-modal (i.e. non-unique solutions) forward model where numerous combinations of interacting parameters could potentially produce identical sequences (Burgess et al., 2006; Burgess and Prince, 2015). Given a specific reef core dataset and *pyReef-Core*, the task of finding the model param-

eters space that best describes the reef core data can be defined as the inverse modelling problem (Jessell, 2002). Mosegaard and Sambridge (2002) highlighted the importance for Monte Carlo methods in analysis of nonlinear inverse problems where no analytical expression for the forward relation between data and model parameters is available. Markov Chain Monte Carlo (MCMC) methods can straight-forwardly quantify uncertainty in model assumptions and parameters (Andrieu et al., 2003). This is particularly useful for SFM approaches (Warrlich et al., 2008) that requires optimisation techniques that lack uncer-

tainty quantification. However, Bayesian inference methods have rarely been applied to reef modelling, despite evidence of



their usefulness when handling models with complex, interrelating parameters (Gallagher et al., 2009). One future development avenue for *pyReef-Core* will consist in integrating a Bayesian inference scheme (MCMC) to provide a methodology for estimation of near-optimal values and uncertainty quantification of free parameters in the model. A useful application of such approach would involve optimisation of environmental threshold and ecological modelling parameters, and then parametrising

of the sediment input and fluid flow boundary conditions based on empirical measurements. Observing cores produced by this method may help in creating 2-D or 3-D conceptualisations of reef stratigraphy.

## 7    Conclusions

Bridging the gap between ecologists and geologists views of coral reef system dynamics is challenging. In this paper, we present *pyReef-Core*, a 1-D deterministic, carbonate SFM that simulates vertical reef sequences comparable to those found in actual

drill cores. The model serves as a basis for investigating the relationship between the key biological processes (i.e. function of coral assemblages interactions based on the GLV equations) involved in coral reef growth, and the influence of changing environmental factors (e.g. sea-level, tectonics, ocean temperature, acidity and nutrient). The significance of the approach lies in its ability to incorporate coral community dynamics into reef growth modelling and understand the responses of coral reefs to environmental disturbances on centennial to millennial timescales at the reef-scale. Exploration of these intermediate scales

is crucial to better understand the enduring growth response of corals in the face of climatic and environmental changes that are expected to have lasting impacts on reefs into the future. As shown in the case studies, generated model predictions cohere well with data and provide a means for explaining observed assemblage patterns. It can help to better constrain the tolerance of shallow-water corals to long-term environmental disturbance, and to quantify the relative dominance of sea-level, tectonics, as well as hydrodynamic energy and sediment input on reef growth.

*Code and data availability.* The source code (written in Python 2.7.6) with examples (Jupyter Notebooks) is archived as a repository on Github and Zenodo (doi:10.5281/zenodo.1080115). The code is licensed under the GNU General Public License v3.0. The easiest way to use *pyReef-Core* is via our Docker container (searching for **pyreef-docker** on Kitematic) which is shipped with the complete list of dependencies and the case studies presented in this paper.

*Competing interests.* The authors declare that they have no conflict of interest.

*Acknowledgements.* T.S. was supported by ARC IH130200012, J.M.W. was supported by ARC DP120101793 and T.S. & J.M.W. were also supported by SREI2020 grants. This research was undertaken with the assistance of resources from the National Computational Infrastructure (NCI), which is supported by the Australian Government and from Artemis HPC Grand Callege supported by the University of Sydney.





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
