# Peer review of "Exploring coral reef responses to millennial scale climatic forcings: insights from the 1-D numerical tool pyReef-Core v1.0"

_Geoscientific Model Development, 2018_

## Referee Comment (RC1) · Anonymous Referee #1 · 9 Apr 2018

This manuscript is well organized and clearly presented. I thought the introductory materials on carbonate systems and environmental controls of reef development (accommodation, hydrodynamic energy, and sediment input) were very concise and adequately thorough. I believe it is essentially ready for publication. My suggestions are for making it more practical and relevant for present and near future coral-reef responses to environmental changes. I also have a few minor editorial suggestions.

The models in this manuscript are for a special circumstance where "populations number. . .is a proxy for carbonate production with larger assemblage population corresponding to faster rate of vertical accretion" (Caption for Figure 6 and elsewhere in

the text). The first sentence of "3 Environmental controls on reef development" (bottom of page 3) states "Coral framework production is strongly linked to biological activity". Actually, there is a general "lack of congruence" between a reef's biological performance (rates of CaCO3 accretion, provision of topographic complexity for shelter and substrata for fish and invertebrates, abundance and diversity of corals) and geological performance (Kleypas et al. 2001). This is because of nearly ubiquitous bioerosion and export of carbonate materials. The authors do acknowledge that their "...model does not consider the destructional processes that occur on the reef due to physical, chemical and biological erosion but does account for erosional process during phases of subaerial exposure..." It is generally true, especially in the Pleistocene and Holocene, that population abundance or living surface cover of corals is associated with overall rate of carbonate production, but because of varying rates of erosion and other destructive forces, rate of carbonate production is only sometimes associated with net rate of vertical growth. I feel the paper does not need to be changed, but I feel the model is limited to particular circumstances.

In the Introduction (line 14), the pyReef-Core model incorportates "coral community dynamics into reef growth modeling at reef-scale resolution". A recent dynamic of reef growth is the combination of disturbances and stressors as the frequencies of disturbances and the duration of stressors both increase. When they both increase, they consistently do not allow the processes of recovery to materialize. Although there are still a number of local coral-reef communities that display remarkably rapid recovery, numerous surveys have indicated that average living coral cover is decreasing circumtropically. This is partly because disturbance is nearly always faster than recovery, damaged or stressed corals generally produce fewer larvae, reef community recovery times become longer when fast-growing branching corals are more vulnerable to stresses and disturbances and are replaced by more tolerant slow-growing corals, combinations of local and global disturbances and stresses result in positive feedbacks that accelerate reef degradation, and degraded reefs decrease the proportion of habitat acceptable to recruiting larvae. In the Holocene, many reefs had time to largely recover

before the next disturbance; otherwise the reefs at the sites would not have developed as well as they had. As more reefs are disturbed and become less favorable for survival of recruits, connectivity is reduced by fewer larvae produced, more areas become unattractive to larvae for settlement, distances between fewer favorable sites become longer, and larval pelagic duration sometimes becomes shorter with increasing temperature. As disturbances become more frequent and do not allow sufficient time for recovery, events become trends and feedback creates ratchets, perpetually inhibiting recovery. Without taking the frequency of disturbances and length of periods of stress, including their overlap, in relation to potential periods of successful recruitment, it does not serve an applied use for the present and near future. However, this is going beyond the scope of this manuscript. I realize that the model in this paper is to reproduce the details of stratigraphic sequences in the past, and it serves this purpose well, so the manuscript should be published.

Minor editorial matters:

Bottom line, page 4 — "acidity" is used to name an environmental factor in oceans. Although the ocean pH may have gotten as low as 7.4 – 7.6 during the Cretaceous and early Paleogene, I don't think the oceans have ever been actually acidic. The term "acidification" is being used to refer to a lowering of pH towards acidity, but it will never actually reach acidity. Second line, page 6 — Change "have a negatively effect" to "have a negative effect". Figure 2 caption — "GLVE equations" is redundant because the E in the initialization is for "equations" First line of text below Table 2 on page 15 – change "data implies" to "data imply" or "data suggest". Top line, page 22 — change "data is scarce" to "data are scarces". Figures 6 and 7 — I have a hard time distinguishing the shades of blue representing different depths. I suspect many readers will have the same problem. Please make the colors more distinct or use different colors for the different depths.

Citations Kleypas JA, Buddemeier RW, Gattuso J-P. 2001. The future of coral reefs in an age of global change. Int J Earth Sciences (Geol Rundsch) 90: 426-437

---

## Referee Comment (RC2) · J. Hill (Referee) · 26 Apr 2018

Exploring coral reef responses to millennial scale climate forcings: insights from a 1-D numerical tool pyReef-Core v1.0 Salles et al

This is a well written paper that details a new 1D stratigraphic forward model specialising in simulating coral reef assemblages over (geologically) short timescales. pyReef-Core contains algorithms that simulate coral reef growth due to changes in water depth, turbidity, flow velocity, wave energy and the assemblage. Assemblages are simulated using a simple L-V type equation set.

[Figure]

pyReef-Core is a unique model in that it is a) open source (rare for SFMs!) and is the first model I know of that attempts to to simulate reef assemblages from an ecological point of view in conjunction with the general controls on reef dynamics. This paper should be published with only minor corrections.

I could access the code (but have not tested it) and I commend the authors for their use of GitHub and Zenodo to archive code.

+++General comments+++

The paper is well written and easy to understand. My only criticism is that the paper contains perhaps too much detail on the controls on carbonate growth which have been well established in the literature for decades (sec 3.1 to 3.3). However, these sections then seem to come to the conclusion we don't know that much, but they are going to be in the model anyway. Perhaps shorter, more succinct summaries with a clear reason for inclusion in the model would clarify this? Another suggestion would be to move the discussion part of these intro sections to the discussion part of the manuscript? I'll leave this to the authors to decide here.

+++Specific comments+++

Pg 1, ln 1: Unclear opening sentence to abstract. Do you mean laterally perpendicular to shore, alongshore or both (in which case, perhaps "spatially" is a better term)? The lateral change and progradation/accretion/retrogression is responsible for the change in core depth: i.e. they are the same thing are they not?

pg 1, ln 5: poorly constrained on centennial to geological timescales, no?

pg 1, ln 6: it doesn't do the inverse though?

pg 2, ln 33: Add Hill et al 2012 as a heuristic tool Hill, Jon, Rachel Wood, Andrew Curtis, and Daniel M. Tetzlaff. 2012. "Preservation of Forcing Signals in Shallow Water Carbonate Sediments." Sedimentary Geology 275-276 (1): 79–92.

pg 3, ln 14: typo \textsc you forgot the \

pg 4: I like this figure - excellent summary

pg 5, ln 6. Does this need a new paragraph?

pg 6, ln 10. So how can you encapsulate this in an algorithm if there's no data? Perhaps some of this needs moving into the discussion? See above general comment.

pg 6, ln 10. Remove sentence: "This objective....". I don't think it adds anything.

pg 7, ln 9-10: As general comment on moving to discussion.

pg 9, ln 15: good explanation of this parameter

pg 11, sec 4.7. 50% is rather arbitrary! Can you give any insight on how the resultant core varies if this is altered to say 25% or 75%? How did you arrive at 50%!?

pg 22, ln 30+: I'm not sure this is relevant here. You don't tackle the inverse problem in this paper and whilst I don't disagree with this at all (as you know!), the linkage to inverse in the abstract and this is tenuous. Perhaps leave removing the inverse and removing the reference to pyReef-Bayes is sufficient here; i.e. you still get to stake out the fact that the inverse problem is what we are trying to solve (as a community), but it's the implication you are doing that in this paper which I don't think sits well.

---

## Author Response (AR1)

**Responses to Reviewers**

*Exploring coral reef responses to millennial scale climatic forcings: insights from a 1-D numerical tool pyReef-Core*

**▬▬▬ Editor's comments to the authors**

**Comment 1:** *Our manuscript preparation guidelines require that all papers must include a model name and version number (or some other unique identifier) in their title. In the submitted version, the version number is missing. Could you please include the version number of the model actually described here, the more since the Zenodo repository already includes two versions.*

**Response:** Following Editor's comment and GMD guidelines, we have added the version number attached to the model described in our manuscript in the title. The version is *v.1.0* and relates to the Zenodo repository (doi:10.5281/zenodo.1080115) included in the code code availability section of the manuscript.

**Comment 2:** *I have furthermore come across one minor thing, that I would like to ask you to correct as well at this stage. Again, in our manuscript preparation guidelines, under "English guidelines and house standards" - "Abbreviations", you will see that the standard abbreviation used in our journal for "years" depends on the meaning (time/date or duration). Throughout the manuscript, I have found "ky" for both duration and dates:*
*○ for durations, these should be corrected to "kyr"*
*○ for dates (e.g., when sea-level stand high, etc.), they should read "ka".*

**Response:** Following Editor's comment, we have corrected the abbreviations for time/date and duration throughout the manuscript.

**Comment 3:** *Finally, and most important, on p. 9, lines 9-10, I read that "The equation requires an initial population for each assemblage $N_i^0$, which is assumed zero for all populations as the basement substrate is unpopulated at the beginning of reef initiation simulations." Although I agree that this translates the fact that the basement substrate is unpopulated initially, it has an unrealistic consequence: it actually means that all the $dN_i/dt$'s from equation (1) are zero initially, so that no populations can develop. Even a single assemblage i that has a zero initial population will be unable to develop according to equation 1. Since pyReef-Core obviously yields non-zero population evolutions as one can see in the results, there must be a special treatment to overcome this barrier.*
*I know that the "all populations are zero" state represents an unstable equilibrium, so I am wondering whether the model relies on some random procedure to start (initial negligibly small populations are sufficient to trigger population growth) or some other solution. The paper needs to provide an accurate description of the actually used procedures adopted in the model. With respect to the starting procedure, it is currently incomplete.*

**Response:** Any population growth from an initial state where no assemblages is present is triggered by the environmental conditions and is set using a 'turn-on' criterion presented in section 4.7. Following Editor's comment, we have added at the end of section 4.2 further explanations regarding the initialisation of coral assemblage population to make it clearer to the readers. In addition, we are now explicitly referring to the 'turn-on' criterion section (section 4.7) in the manuscript where the reasoning behind the procedure is explained in more details.

**▬▬▬ First reviewer's comments to the authors**

**General Comment:** *This manuscript is well organized and clearly presented. I thought the introductory materials on carbonate systems and environmental controls of reef development (accommodation, hydrodynamic energy, and sediment input) were very concise and adequately thorough. I believe it is essentially ready for publication. My suggestions are for making it more practical and relevant for present and near future coral-reef responses to environmental changes. I also have a few minor editorial suggestions.*

*The models in this manuscript are for a special circumstance where "populations number. . .is a proxy for carbonate*

*production with larger assemblage population corresponding to faster rate of vertical accretion" (Caption for Figure 6 and elsewhere in the text). The first sentence of "3 Environmental controls on reef development" (bottom of page 3) states "Coral framework production is strongly linked to biological activity". Actually, there is a general "lack of congruence" between a reef's biological performance (rates of CaCO3 accretion, provision of topographic complexity for shelter and substrata for fish and invertebrates, abundance and diversity of corals) and geological performance (Kleypas et al. 2001). This is because of nearly ubiquitous bioerosion and export of carbonate materials. The authors do acknowledge that their". . .model does not consider the destructional processes that occur on the reef due to physical, chemical and biological erosion but does account for erosional process during phases of subaerial exposure. . ." It is generally true, especially in the Pleistocene and Holocene, that population abundance or living surface cover of corals is associated with overall rate of carbonate production, but because of varying rates of erosion and other destructive forces, rate of carbonate production is only sometimes associated with net rate of vertical growth. I feel the paper does not need to be changed, but I feel the model is limited to particular circumstances.*

*In the Introduction (line 14), the pyReef-Core model incorportates "coral community dynamics into reef growth modeling at reef-scale resolution". A recent dynamic of reef growth is the combination of disturbances and stressors as the frequencies of disturbances and the duration of stressors both increase. When they both increase, they consistently do not allow the processes of recovery to materialize. Although there are still a number of local coral-reef communities that display remarkably rapid recovery, numerous surveys have indicated that average living coral cover is decreasing circumtropically. This is partly because disturbance is nearly always faster than recovery, damaged or stressed corals generally produce fewer larvae, reef community recovery times become longer when fast-growing branching corals are more vulnerable to stresses and disturbances and are replaced by more tolerant slow-growing corals, combinations of local and global disturbances and stresses result in positive feedbacks that accelerate reef degradation, and degraded reefs decrease the proportion of habitat acceptable to recruiting larvae. In the Holocene, many reefs had time to largely recover before the next disturbance; otherwise the reefs at the sites would not have developed as well as they had. As more reefs are disturbed and become less favorable for survival of recruits, connectivity is reduced by fewer larvae produced, more areas become unattractive to larvae for settlement, distances between fewer favorable sites become longer, and larval pelagic duration sometimes becomes shorter with increasing temperature. As disturbances become more frequent and do not allow sufficient time for recovery, events become trends and feedback creates ratchets, perpetually inhibiting recovery. Without taking the frequency of disturbances and length of periods of stress, including their overlap, in relation to potential periods of successful recruitment, it does not serve an applied use for the present and near future. However, this is going beyond the scope of this manuscript. I realize that the model in this paper is to reproduce the details of stratigraphic sequences in the past, and it serves this purpose well, so the manuscript should be published.*

**Response:** We would like to thank the reviewer for raising these 2 comments one on carbonate framework production *vs* biological performance or growth and the other about disturbance. We acknowledge and value the review comments about the general "lack of congruence" between a reef's biological performance and geological performance, but agree with the review that it is out of scope of the current MS. However in future iterations of the model, we plan to implement a modified version of the coral production frame- work so as to better investigate the case where vertical accretion rate do not reflect the growth rate of corals.

In regards to the second comment, we agree that accounting for the frequency of disturbances and length of periods of stress, is critical if one wants to investigate present and near future reef evolution processes. Adding this in our modelling framework might be possible by incorporating some sort of recruitments function as suggested by the referee.

**Comment 1:** *Bottom line, page 4 – "acidity" is used to name an environmental factor in oceans. Although the ocean pH may have gotten as low as 7.4 – 7.6 during the Cretaceous and early Paleogene, I don't think the oceans have ever been actually acidic. The term "acidification" is being used to refer to a lowering of pH towards acidity, but it will never actually reach acidity.*

**Response:** Following reviewer's comment, we have modified the text to prevent any confusion and we are now referring to pH instead of acidity throughout the manuscript.

**Comment 2:** *Second line, page 6 – Change "have a negatively effect" to "have a negative effect".*

**Response:** we have changed the corresponding line as suggested by the reviewer.

**Comment 3:** *Figure 2 caption – "GLVE equations" is redundant because the E in the initialization is for "equations".*

**Response:** We have delete "equations" to avoid redundancy.

**Comment 4:** *First line of text below Table 2 on page 15 – change "data implies" to "data imply" or "data suggest".*

**Response:** We have changed the text to "data suggest" as proposed by the reviewer.

**Comment 5:** *Top line, page 22 – change "data is scarce" to "data are scarce".*

**Response:** We have changed the text to "data are scarce".

**Comment 6:** *Figures 6 and 7 — I have a hard time distinguishing the shades of blue representing different depths. I suspect many readers will have the same problem. Please make the colors more distinct or use different colors for the different depths.*

**Response:** Following reviewer's comment, we have modified the colours to make them easier to read. To keep consistent over the manuscript we have modified the colours in figures 2, 5, 6, 7 and 8.

**▬▬▬▬  Second reviewer's comments to the authors**

**General Comment:** *The paper is well written and easy to understand. My only criticism is that the paper contains perhaps too much detail on the controls on carbonate growth which have been well established in the literature for decades (sec 3.1 to 3.3). However, these sections then seem to come to the conclusion we don't know that much, but they are going to be in the model anyway. Perhaps shorter, more succinct summaries with a clear reason for inclusion in the model would clarify this? Another suggestion would be to move the discussion part of these intro sections to the discussion part of the manuscript? I'll leave this to the authors to decide here.*

**Response:** We would like to thank Jon Hill for his insightful comments on the paper, as these comments led us to an improvement of the work. Our revisions reflect all reviewer' suggestions and comments. Detailed responses to reviewers are given below. An update version of the manuscript is available as a pdf as a supplement.

Following reviewer's comment we have shorten by half (from 3 to 1.5 pages) the literature part on environmental controls (subsections 3.1 to 3.3). We believe these 3 sections are more useful before the part describing pyReef-Core (section 4) than in the discussion as they put in perspective the different parameters that the model intents to simulate and also highlight the main forcing conditions that drive coral assemblages evolution in our 1D framework. We have modified the last section (3.3) so that we acknowledge more the work done in respect to the control of sediment input on coral communities evolution. The fact that some of the effects of these environmental conditions are still unknown is the main reason why we should aim to try to simulate them in order to gain some insights from numerical models and potentially improve our general understanding of the complex interactions between corals and their environments.

**Comment 1:** *Pg 1, ln 1: Unclear opening sentence to abstract. Do you mean laterally perpendicular to shore, alongshore or both (in which case, perhaps "spatially" is a better term)? The lateral change and progradation/accretion/retrogression is responsible for the change in core depth: i.e. they are the same thing are they not?*

**Response:** Following reviewer's comment, we have modified the first sentence and used the term spatially as suggested by the reviewer.

**Comment 2:** *Pg 1, ln 5: poorly constrained on centennial to geological timescales, no?*

**Response:** We have changed millennial to geological timescales.

**Comment 3:** *Pg 1, ln 6: it doesn't do the inverse though?*

**Response:** Following reviewer's comment, we have remove the term inverse in the sentence as it was misleading.

**Comment 4:** *Pg 2, ln 33: Add Hill et al 2012 as a heuristic tool Hill, Jon, Rachel Wood, Andrew Curtis, and Daniel M. Tetzlaff. 2012. "Preservation of Forcing Signals in Shallow Water Carbonate Sediments." Sedimentary*

*Geology 275-276 (1): 79–92.*

**Response:** We have added the reference to Hill et al., 2012.

**Comment 5:** *Pg 3, ln 14: typo ʏou forgot the*

**Response:** We have corrected the typo.

**Comment 6:** *Pg 5, ln 6. Does this need a new paragraph?*

**Response:** We have merged the 2 paragraphs together.

**Comment 7:** *Pg 6, ln 10. So how can you encapsulate this in an algorithm if there's no data? Perhaps some of this needs moving into the discussion? See above general comment.*

**Response:** This point has been addressed now, as explained in our response to the general comment.

**Comment 8:** *Pg 6, ln 10. Remove sentence: "This objective....". I don't think it adds anything.*

**Response:** Following reviewer's suggestion, we have removed the sentence.

**Comment 9:** *Pg 7, ln 9-10: As general comment on moving to discussion.*

**Response:** This point has been addressed now, as explained in our response to the general comment.

**Comment 10:** *Pg 11, sec 4.7. 50% is rather arbitrary! Can you give any insight on how the resultant core varies if this is altered to say 25% or 75%? How did you arrive at 50%!?*

**Response:** First we would like to state that this parameter is user-defined in pyReef-Core and can be changed in the XmL file by adjusting the **facOpt** parameter (http link). Following reviewer's comment, we have modified the section 4.7 and added a new sentence at the end to reflect the fact that this parameter can be set by the user. The change in 'turn-on' criteria value will result in different evolution and therefore can change significantly the resulting core. This is especially true for simulations in which assemblage population number fall to 0 due for example to reef drowning or aerial exposure. Difference between a value of 25% compare to 75% will enable assemblages to start growing even though their optimal environmental conditions are not reached. A high value like 75% could result in simulations in which particular coral assemblages will never be able to grow due to more restrictive environmental conditions.

**Comment 11:** *Pg 22, ln 30+: I'm not sure this is relevant here. You don't tackle the inverse problem in this paper and whilst I don't disagree with this at all (as you know!), the linkage to inverse in the abstract and this is tenuous. Perhaps leave removing the inverse and removing the reference to pyReef-Bayes is sufficient here; i.e. you still get to stake out the fact that the inverse problem is what we are trying to solve (as a community), but it's the implication you are doing that in this paper which I don't think sits well.*

**Response:** Following reviewer's suggestions, we have removed *inverse* in the abstract as well as the reference to pyReef-Bayes from this section, the inverse problem and the MCMC approach is indeed not relevant to the work described in this paper and will be published later.

[revised manuscript text omitted]